# Accounting for Greenhouse Gas Emissions from Forest Edge Degradation: Gold Mining in Guyana as a Case Study

**Sandra Brown [1],[†], Abu R. J. Mahmood [2],[3],[*],[‡], Katherine M. Goslee [1], Timothy R. H. Pearson [1], Hansrajie Sukhdeo [4], Daniel N. M. Donoghue [2] and Pete Watt [5]**

1    Winrock International, Arlington, VA 22202, USA; kgoslee@winrock.org (K.M.G.);
      tpearson@winrock.org (T.R.H.P.)
2    Department of Geography, Durham University, Durham DH1 3LE, UK; danny.donoghue@durham.ac.uk
3    Institute of Forestry and Environmental Sciences, Chittagong University, Chattogram 4331, Bangladesh
4    Guyana Forestry Commission, Georgetown 00592, Guyana; hans.sukhdeo@gmail.com
5    Indufor Asia-Pacific, Auckland City 1147, New Zealand; pete.watt@indufor-ap.com
*    Correspondence: ifescu@gmail.com
†    Deceased.
‡    Currently working for the Food and Agriculture Organization of the United Nations based in Asia-Pacific.

**Abstract:** *Background and Methods:* Degradation of forests in developing countries results from multiple activities and is perceived to be a key source of greenhouse gas emissions, yet there are not reliable methodologies to measure and monitor emissions from all degrading activities. Therefore, there is limited knowledge of the actual extent of emissions from forest degradation. Degradation can be either in the forest interior, with a repeatable defined pattern within areas of forest, as with timber harvest, or on the forest edge and immediately bounding areas of deforestation. Forest edge degradation is especially challenging to capture with remote sensing or to predict from proxy factors. This paper addresses forest edge degradation and: (1) proposes a low cost methodology for assessing forest edge degradation surrounding deforestation; (2) using the method, provides estimates of gross carbon emissions from forest degradation surrounding and caused by alluvial mining in Guyana, and (3) compares emissions from mining degradation with other sources of forest greenhouse gas emissions. To estimate carbon emissions from forest degradation associated with mining in Guyana, 100 m buffers were located around polygons pre-mapped as mining deforestation, and within these buffers rectangular transects were established. Researchers collected ground data to produce estimates of the biomass damaged as a result of mining activities to apply to the buffer area around the mining deforestation. *Results:* The proposed method to estimate emissions from forest edge degradation was successfully piloted in Guyana, where 61% of the transects lost 10 Mg C ha$^{-1}$ or less in trees from mining damage and 46% of these transects lost 1 Mg C ha$^{-1}$ or less. Seventy percent of the damaged stems and 60% of carbon loss occurred in the first 50 m of the transects. The median loss in carbon stock from mining damage was 2.2 Mg C ha$^{-1}$ (95% confidence interval: 0.0–10.2 Mg C ha$^{-1}$). The carbon loss from mining degradation represented 1.0% of mean total aboveground carbon stocks, with emissions from mining degradation equivalent to ~2% of all emissions from forest change in Guyana. *Conclusions:* Gross carbon emissions from forest degradation around mining sites are of little significance regardless of persistence and potential forest recovery. The development of cost- and time-effective buffers around deforestation provides a sound approach to estimating carbon emissions from forest degradation adjacent to deforestation including surrounding mining. This simple approach provides a low-cost method that can be replicated anywhere to derive forest degradation estimates.

**Keywords:** forest degradation; mining; REDD+; greenhouse gas emissions

## 1. Introduction

International emission reduction programs (especially Reducing Emissions from Deforestation and Degradation, Conservation of Forest Carbon Stocks, Sustainable Management of Forests and Enhancement of Forest Carbon Stocks (REDD+)) have principally focused on the reporting of deforestation activities, as robust and well-tested methods and data are available for monitoring such changes across forested landscapes. In contrast, although degradation of forests in developing countries is perceived to be important for global greenhouse gas emissions (GHG) [1], full inclusion of forest degradation accounting is very rare due to lack of methods and excessive costs of available approaches.

Most REDD+ financing mechanisms and protocols require emissions from forest degradation to be accounted for when "significant", defined by the World Bank under its Carbon Fund as more than 10% of all forest-related emissions [2] and some works have also been published that highlight a critical degradation threshold [3–5]. Identifying which activities cause "significant" forest degradation in any given country requires cost-effective measurement of their emissions, yet measuring and monitoring changes in carbon stocks (emissions and removals) due to forest degradation is decidedly more complex and costly than measuring and monitoring carbon emissions due to deforestation [1,6]. To develop a measurement, reporting, and verification (MRV) system for forest degradation, it is necessary that the causes of degradation be considered and the likely impact on carbon stocks be assessed. Examples of activities that commonly result in forest degradation include legal and illegal logging, human-set fires that escape into the forest, fuelwood collection, and persistent livestock grazing.

Forest degradation refers to losses in forest cover that do not qualify as deforestation. Such degradation can be caused by many different drivers, but we argue that it consists of two different patterns of changes in forest cover. Forest degradation can occur on the edge of the forest, caused by drivers inadvertently or opportunistically taking advantage of access from deforested areas. Such degradation typically has a limited depth of penetration into the surrounding forest but is spread across an area with a variable and unpredictable density; examples include human and environmental impacts surrounding mining, livestock encroachment into forests, anthropogenically-set fires, and fuelwood collection. The second form of forest degradation occurs deep in the interior of the forest with the specific purpose (advertent or inadvertent) of an activity that lowers the forest carbon stock. This form of forest degradation impacts a defined area in a recognizable pattern and with a close relationship to a statistic that can provide activity data; examples include timber harvest (with activity data of timber volumes extracted), or low-grading for understory crop production (with activity data of plantation area). We term the second form of forest degradation "forest interior degradation" and the first form "forest edge degradation". Here, we focus on how to account for greenhouse gas emissions from forest edge degradation as exemplified by forests adjacent to alluvial gold mines in Guyana.

Mining for gold and other precious materials is a common cause of deforestation in many regions around the world [7], including many countries in South America [8,9], Southeast Asia [10,11] and parts of Africa [12–14]. As a result, many forested landscapes around the world have been severely altered [15]. The elevated rate of deforestation from gold-mining has been observed in the recent past due to multiple factors, including an increase in gold prices in the international market [16], an increase in accessibility to mines, lack of formalization of artisanal and small-scale mining [17], and the expectation of high short-term earnings from mining and selling gold [18]. Swenson et al. [19] and Asner et al. [9] showed mining activities in South America were both regulated and unregulated, primarily artisanal, and more often illegal than legal.

It is highly likely that forests surrounding mining areas are degraded due to the nature of mining activity, including tree removal during pre-mining exploration and for timber for camp construction and fuel for cooking, as well as post-mining impacts such as toxic runoff and flooding as a result of mine tailings. Assessing deforestation from mining is relatively simple as remote sensing can be routinely used to record areas that previously had forest cover. However, the areas associated with forest degradation resulting from mining and the consequent GHG emissions are unknown. This paper is focused on identifying a viable approach to quantify the impact of forest edge degradation as exemplified by the degradation of forests surrounding mining sites. Here, mining can represent any forest edge degradation surrounding deforestation such as livestock encroachment or the penetration of desiccating climate on deforestation edges.

Two basic elements are needed to estimate GHG emissions associated with forest degradation: activity data and emission factors. Activity data (AD) refer to the quantity of an activity that results in GHG emissions, such as the area of land degraded. Emission factors (EF) are the estimated amount of emissions of GHGs per unit of activity, such as megagrams of carbon emitted per hectare (Mg C ha$^{-1}$) degraded [6,20]. EF combined with AD gives an estimate of total gross emissions from the activity. Available studies show how much uncertainty in the estimation of AD and EF is associated with forest degradation [6].

A method for estimating gross emissions from forest interior degradation in the form of selective logging in tropical forests has been developed by Pearson et al. [21,22] using the Intergovernmental Panel on Climate Change (IPCC) gain–loss approach [23]. This approach focuses on the direct change in carbon stocks and therefore requires the measurement of tree mortality and damage rather than an estimation of the difference in carbon stocks before and after a degrading event. This method has been found to be more appropriate for estimating the impact of degrading activities, especially when carbon stocks are variable across the forest and the carbon loss is relatively small. In such situations, the gain–loss approach requires fewer measurements to reach a reasonable level of certainty than would be required by measuring carbon stocks before and after damage. However, this interior form of forest degradation is not directly associated with deforestation, and incorporates recognizable patches of cleared forest (in roads, log pile decks, temporary camps, and tree fall gaps), and has readily available activity data in extracted timber volumes. In contrast, degradation around mining results from limited and diffuse incidental tree mortality of predominantly small diameter trees. As such it is complex and difficult to capture accurately in remote sensing and the only reliable activity data are the length of the deforestation edge.

Several studies have used remote sensing data for mapping and estimation of deforestation from mining and the majority of these studies were in South America [9,19,24–27]. A range of remote sensing data, both from active (LiDAR) [25] and passive (MODIS, Landsat, IRS LISS III, WorldView, Digital Globe) [16,24,26] sensors, have been used in these studies. These studies have demonstrated that areas of deforestation due to artisanal mining are visible from space and could easily be mapped at a regional or national-scale using a combination of remote sensing and limited ground-survey methods [9]. No studies have investigated the extent of forest degradation and associated GHG emissions from artisanal or small-scale mining.

The purpose of our study was to analyze forest edge degradation, based on a case study in Guyana, where gold mining is the main cause of deforestation (more than 88% of total deforestation [28]; Figure 1), and to arrive at a first estimation of carbon emissions, both in absolute terms and relative to deforestation and other forms of forest degradation. In this process, this paper seeks to develop and test methods for estimating gross carbon emissions caused by forest edge degradation surrounding areas of deforestation (as exemplified by mining).

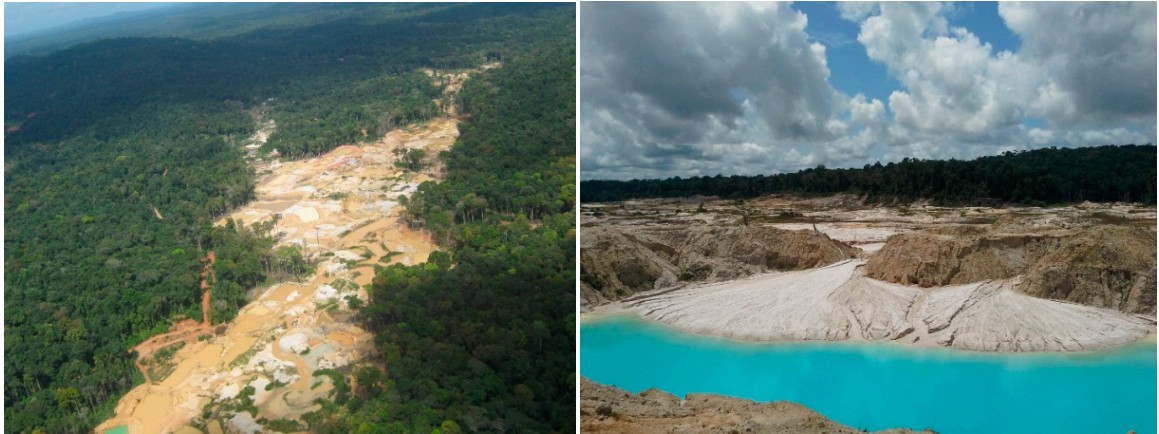

**Figure 1.** An example of mining activities inside the forest near Mahdia, known as one of the hot spots for alluvial gold mining in Guyana. Gold mining is one of the key drivers of carbon emissions through deforestation and forest degradation. The photo on the left shows the aerial view and the photo on the right illustrates the ground conditions of a typical mining site in Guyana.

In this paper, we: (1) provide estimates of the total gross carbon emissions from forest edge degradation adjacent to mining areas, (2) compare the carbon emissions from forest degradation with carbon emissions from deforestation both associated with mining deforestation emissions and total forest emissions in Guyana, and (3) provide a sound method that can be used for estimating carbon emissions from forest edge degradation surrounding areas of deforestation.

## 2. Materials and Methods

The study was conducted in the country of Guyana, on the northeastern coast of South America. The land area of Guyana is approximately 21 million hectares, of which approximately 18 million hectares are forest land. The population of Guyana is approximately 782,000, with 90% residing on the coastal strip of the country. Alluvial gold mining is one of the primary economic activities in Guyana. As mapped in the forest change maps (2011–2015) of the Guyana Forestry Commission, the majority of the mining sites are found in the northern and central parts of the country. [29] Based on a geographic information system (GIS) assessment, this study selected the region of Mahdia as the study site based on the representation of mining practices and sizes of mining operations. All mining sites examined for this research were between latitudes 4.616° N and 5.473° N and longitudes 58.395° W and 59.612° W.

To define the area of forest degradation (AD) that results from mining activity, we developed a 100 m buffer around all identified deforestation areas that exceeded 1 ha in Guyana's forest monitoring system for the years between 2010 and 2015 using a geographic information system (GIS). The 100 m buffer reflected initial analysis across 500 m buffers that showed that minimal to no degradation occurred beyond the initial 100 m. The mining degradation activity data were estimated using the area of the 100 m buffer zones. The approach was chosen as a pragmatic and cost-effective method to develop reliable activity data.

To estimate the change in forest carbon stocks, field data were collected from transects (Figure 2). Transects were located across the width of the 100 m buffer.

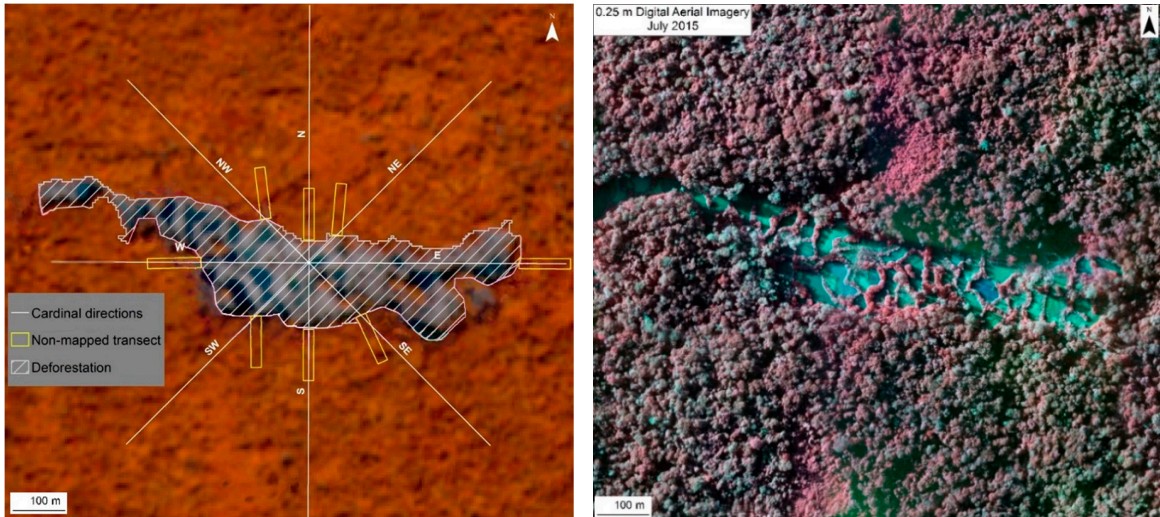

**Figure 2.** Example of a mined area, with pre-selected locations for degradation transects (left). The rectangles outlined in yellow show the potential locations of 20 by 100 m transects established within the fixed-area 100 m buffer, 5 m RapidEye (left) and 0.25 m digital aerial infra-red imagery (right) have been used by the Guyana Forestry Commission for mapping and validation, respectively.

*2.1. Data Sets*

The study used remote sensing datasets spanning from 2010 to 2015, and field data collected between July and August 2015 (Table 1).

**Table 1.** Description of the datasets collected from Guyana Forestry Commission and from the field surveys, used for the mining degradation study. The years in the parenthesis indicate the assessment periods indicated in the Guyana Monitoring, Reporting, and Verification System (MRVS) project.

| Data Type | Data Products | | Period | Data Source |
|---|---|---|---|---|
| **Vector** | Guyana forest change map 2011 (year 2) | | October 2010–December 2011 | Guyana Forestry Commission |
| | Guyana forest change map 2012 (year 3) | | January 2012–December 2012 | |
| | Guyana forest change map 2013 (year 4) | | January 2013–December 2013 | |
| **Raster** | 5 m-RapidEye Satellite Imagery | | 2011–2014 | |
| **Ground survey** | 41 rectangular transects of 20 m × 100 m (0.2 ha) each | | July–August 2015 | Primary Data |
| | Year | No. of transects | | |
| | 2011 | 11 | | |
| | 2012 | 16 | | |
| | 2013 | 14 | | |
| | Total | 41 | | |

*2.2. Remote Sensing Sampling Design*

The remote sensing sampling design applied high resolution (5 m) RapidEye multispectral satellite imagery. The location of potential sampling sites was established from the multitemporal RapidEye imagery (2011–2014) that had been used to identify and map deforestation by driver. First, deforestation from mining activity was identified from RapidEye imagery as forest clearings with sharp boundaries, often in linear clusters in remote areas and near water [30]. All such areas that exceeded one hectare were mapped as deforestation from mining. The accuracy of the annual deforestation mapping products was independently assessed using an aerial multispectral imaging system that

captures images across Guyana with resolution ranging from 25 to 60 cm [31]. The assessment shows high accuracy for deforestation (~99%).

The 100 m wide buffers were established around identified mining deforestation. Field sites were pre-selected to ensure representation of mining practices and sizes of mining operations across the country. We identified five testing sites and installed transects at these sites all with the same dimensions of 20 m (width) × 100 m (length).

Figure 2 shows the sampling approach. Transect locations were pre-established using forest change maps of three time periods: 2011–2012, 2012–2013, and 2013–2014 (Table 1), produced by the Guyana Forestry Commission. Areas of deforestation from mining during these time periods were identified with a representative distribution across the landscape using various information sources including an updated road map, a road layer from Google Earth Pro, and elevation data from Shuttle Radar Topography Mission (SRTM). The locations for an initial set of transects were randomly selected from the identified areas of mining deforestation, then 41 transect locations were randomly selected from this initial set of transects. For each selected deforestation polygon centroid, out of eight cardinal points, two were chosen at random (Figure 2). All potential transect locations were established on maps in advance of the ground surveys.

*2.3. Field Data Collection*

Using the gain–loss approach to identify carbon loss resulting from human activity, measurements focused on trees (or stumps) whose mortality was caused by human impact, such as harvesting to build a mining camp or a trail, or mortality that resulted from flooding or mine tailings.

Across all transects, the diameter at breast height (DBH) of all trees ≥ 10 cm DBH was measured. For all damaged or killed trees (lying or standing), the following data were recorded: the species; DBH where ≥10 cm when available or basal diameter at 5 cm above ground, which was then converted to DBH using the taper factor shown in Equation 1; distance from start of transect to the damaged tree; whether damage was likely due to natural causes or was human-caused (e.g., cutting, snapped, broken, washed out roots, root burial by sediments, trail construction, flooding, or presence of toxic mining waste). In the tropical forests, measuring tree height is difficult and expensive [32]. In this study, tree height was not included due to the possibility of systematic measurement error, time and budget constraints, and inaccessible ground conditions resulting from mining. The systematic errors include measurement error due to dense canopy cover, measurement error due to the presence of tall trees, and irregular crown condition [33]. In each transect, evidence of degrading activities was also recorded with a description of the disturbance and measurement of distance from start of transect.

The number of transects in each site established by plot type is given in Table 1. The number of transects varied for each year of deforestation due to availability. Available time and resources initially allowed for measurement of a total of 41 transects, and upon data analysis it was found that this number met the threshold of <20% uncertainty at a 95% confidence interval. Throughout data collection and data processing, independent checking and correction were conducted as integral parts of quality control and quality assurance.

*2.4. Data Analysis*

2.4.1. Estimation of Carbon Stocks in Aboveground Biomass

Tree data were converted to aboveground biomass using an equation from Chave et al. [34] that estimates biomass for moist forests using DBH and species-specific wood density (this equation has been tested and used for estimating aboveground biomass in Guyana's REDD+ forest carbon monitoring system and is used here to remain in keeping with that system). Estimates of belowground biomass were not included in this analysis and comparisons were made to the aboveground biomass emissions from other sources only. Where the DBH was measured directly this was used in the Chave et al. [34] equation. However, if the tree was removed, the diameter of the stump at 5 cm above

ground was measured and the DBH was estimated from the basal diameter using a taper factor ($T_{taper}$) (see Equation (1)):

$$D_{estimated} = D_{base} - \left[ \left\{ 1.3 - \left( \frac{H_{base}}{100} \right) \right\} \times T_{taper} \right] \qquad (1)$$

where:

$D_{estimated}$ = estimated diameter at breast height, cm;

$D_{base}$ = basal diameter, taken at 5 cm above the ground, cm;

$H_{base}$ = height of the basal diameter measurement, 5 cm;

$T_{taper}$ = variation of unit of diameter over a unit of length, 0.79 cm m$^{-1}$, as derived for and used in Guyana's forest inventory.

DBH was measured for 98% of the trees recorded and basal diameter was measured for 22%.

Aboveground biomass was converted to megagrams per hectare using a scaling factor. This was multiplied by the carbon fraction 0.47 [35] to convert to Mg C ha$^{-1}$.

### 2.4.2. Statistical Analysis

The total carbon content in live trees across 41 transects showed a normal distribution, whereas the carbon loss values (Mg C ha$^{-1}$) associated with mining-related tree mortality across 41 transects was skewed and non-normally distributed [36,37]. Further transformation of the carbon loss values using logarithmic, square root and reciprocal approaches did not satisfy the assumption of normality. Outliers in the values of carbon loss (Mg C ha$^{-1}$) resulted in the median value serving as a better descriptor of the typical carbon emissions than the mean value. Therefore, non-parametric statistical analyses-median, the Wilcoxon rank-sum test [38], and Mann and Whitney's U statistic [39] were applied to conduct statistical analysis using the values of carbon loss (Mg C ha$^{-1}$) from 41 transects. Of note is that the field measurements represent cumulative damage between 2011 to 2015 with many of the deforestation sites first mapped in 2011. The Wilcoxon rank-sum test was run between the transects separated by year to investigate whether carbon emissions due to mining differ significantly between years.

An evaluation of uncertainty (confidence interval—CI) associated with a sample-based point estimate using the percentile ($p$) method [40] is needed to relate to the corresponding population parameter. For the evaluation, we generated 10,000 bootstrap samples drawn within each of the 10,000 replications of a bootstrap simulation, which is ten times more replications than recommended by Efron and Tibshirane [41]. The 10,000 bootstrap samples were then used to derive three different CIs following three different nonparametric bootstrap resampling procedures: the bootstrap normal (BN) method, the bootstrap percentile (BP) method, and the bootstrap bias-corrected and accelerated (BCa) method [42]. Finally, the sample-based uncertainty (CI) estimate of carbon loss was evaluated against the uncertainty (CI) values derived from those three different nonparametric bootstrap resampling procedures. The standard error and bias were calculated using the bootstrap resampling method [43,44].

### 3. Results

Two thousand nine hundred and forty-five (2945) tree stems were recorded in the 41 transects. Live trees comprised 92.1% of all stems with tree mortality due to mining activities comprising 3.8% of stems and tree mortality due to natural causes comprising the remaining 4.1% of stems. No drivers of forest loss other than mining and natural disturbances had been observed during the field visits, mainly due to remoteness and inaccessibility of the transect locations.

### 3.1. Change in Carbon Stocks

The carbon loss due to mining-induced tree damage varied significantly, ranging from 0.0 to 96.7 Mg C ha$^{-1}$ of the sampled transect (Table 2). The median loss in carbon stock across all years was 2.2 Mg C ha$^{-1}$ (0.0–10.2 Mg C ha$^{-1}$ at 95% CI based on the BCa method). This represents a mining degradation emission factor of 8.0 Mg CO$_2$ ha$^{-1}$ to be applied across the total calculated areas of buffer around mining deforestation.

**Table 2.** Summary statistics of carbon loss by year in mining damaged trees, in Mg C ha$^{-1}$ in 41 transects. The "year" column indicates the period of mapping during which mining activities were identified by Guyana Forestry Commission mapping (2011–2013). The uncertainty range at 95% confidence interval was estimated using percentile method (*p*), bootstrap normal method (BN), bootstrap percentile method (BP), and bootstrap bias-corrected and accelerated method (BCa), respectively. The bootstrapped values were derived from the resampling through 10,000 iterations.

| Year | # of Transects | Min | Max | Mean | Median | Bootstrap Bias | Bootstrap Standard Error | 95% Confidence Interval | |
|------|----------------|-----|-----|------|--------|----------------|--------------------------|-------|-------|
| | | | | | | | | Lower | Upper |
| 2011 | 11 | 0.0 | 57.1 | 10.5 | 4.3 | 1.0 | 4.8 | 0.0<br>−5.2<br>0.0<br>0.2 | 16.1 (*p*)<br>13.8 (BN)<br>15.2 (BP)<br>18.3 (BCa) |
| 2012 | 16 | 0.0 | 70.3 | 10.1 | 0.9 | 0.7 | 2.2 | 0.0<br>−3.5<br>0.0<br>0.0 | 8.4 (*p*)<br>5.2 (BN)<br>7.3 (BP)<br>8.5 (BCa) |
| 2013 | 14 | 0.0 | 96.7 | 16.3 | 8.5 | −0.9 | 5.4 | 0.0<br>−2.1<br>0.0<br>0.0 | 19.0 (*p*)<br>19.2 (BN)<br>16.7 (BP)<br>16.0 (BCa) |
| Total | 41 | 0.0 | 96.7 | 12.3 | 2.2 | 0.8 | 2.8 | 0.0<br>−3.4<br>0.0<br>0.0 | 10.2 (*p*)<br>7.8 (BN)<br>10.2 (BP)<br>10.2 (BCa) |

Despite the time span over which degradation had occurred (2 to 4 years), no significant differences were observed in the carbon loss values by year (Table 3). Approximately 61% of transects had 10 Mg C ha$^{-1}$ or less of damage, with the highest frequency (46.3%) seen in the <1 Mg C ha$^{-1}$ class (Figure 3).

**Table 3.** Results of the Wilcoxon rank-sum test [38] used for testing the null hypothesis of carbon loss from mining, indicating that there is no difference in carbon loss from mining across the years 2011, 2012, and 2013.

| Year | # of Transects | Z Value | Probability (*p*) Value [a] |
|------|----------------|---------|------------------------------|
| 2011<br>2012 | 11<br>16 | −0.684 | 0.494 (0.577) |
| 2011<br>2013 | 11<br>14 | −0.139 | 0.889 (0.484) |
| 2012<br>2013 | 16<br>14 | −0.838 | 0.402 (0.413) |

[a] Values in parentheses are the probability that the transects in first year are larger than the transects in the second year.

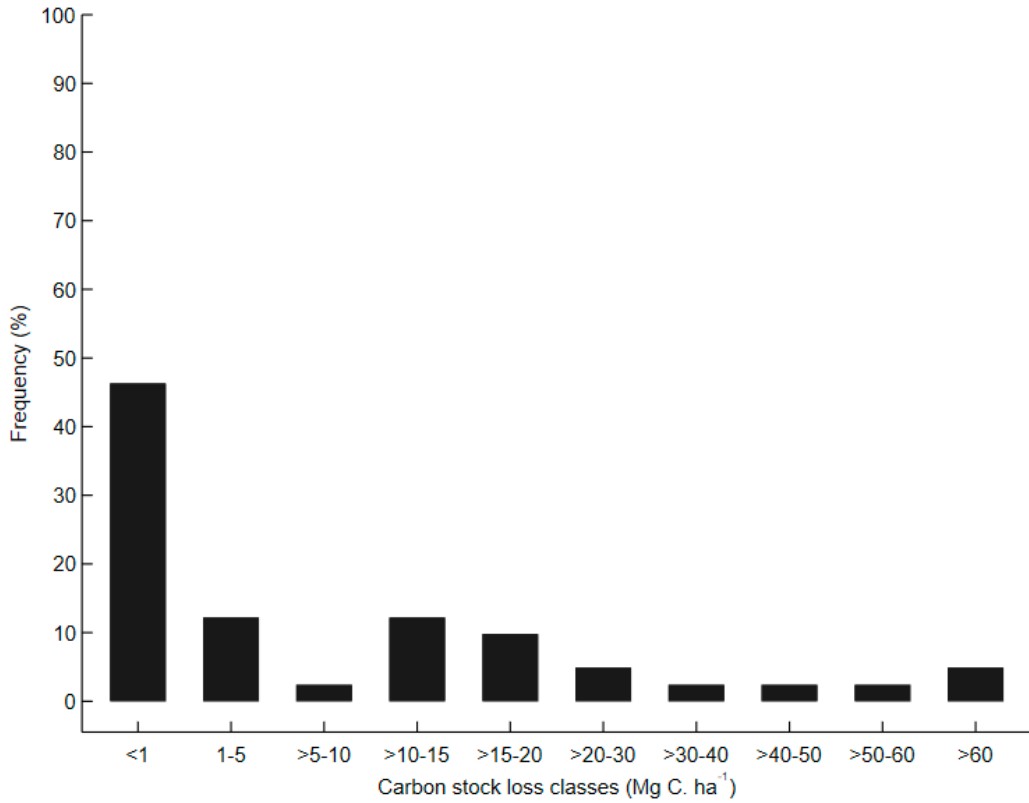

**Figure 3.** Frequency distribution of the carbon stock loss (Mg C. ha$^{-1}$) from trees measured from 41 rectangular transects damaged by mining.

Carbon loss due to mining damage represented only 1.2% of live tree biomass and was estimated to be lower than damage due to natural causes (equivalent to 20% of the losses to natural causes; Table 4). The calculated annual rate of mortality was approximately 0.3% for mining damage and 1.6% for other causes (estimated as the amount of carbon emissions divided by the aboveground carbon stock, expressed as a percent, and assuming an average period of 4 years).

**Table 4.** The mean carbon content in all live trees and the median carbon content in mining damaged trees and in other damaged trees, all in units of Mg C ha$^{-1}$, by year from 41 transects.

| Year | Live Tree [a] | Mining Damage [b] | Other Damage [b] | Total Aboveground [a] Biomass (Live Biomass + Biomass of All Damage) |
|---|---|---|---|---|
| **2011** | 201.0 | 4.3 | 12.1 | 230.0 |
| **2012** | 188.8 | 0.9 | 12.8 | 215.0 |
| **2013** | 183.2 | 8.5 | 8.1 | 211.8 |
| **All** | 190.2 (±27.6) | 2.2 (0.0–10.2) | 11.6 (1.6–17.5) | 218.0 (±29.4) |

[a] values are mean ± with 95% CI value for live trees and for total aboveground biomass; [b] median for mining and other damaged trees with lower and upper bounds for 95% CI.

To address the impact of the distance from the mining locus on forest degradation, we plotted the distribution of the number of mining damaged stems and the associated carbon loss at 10 m intervals as a percent of all mining damaged trees and total carbon loss along the transect length (Figure 4), starting from the edge of a deforestation polygon. The distribution of mining damaged stems shows an expected gradual decline with an increase in distance from the edge of the deforestation polygon (Figure 4). In the 41 transects, 70% of the mining damaged stems were located in the first half (0–50 m)

of the transects with 60% of the carbon loss occurring in the same distance. Within 80 m from the deforestation edge, 88% of the damaged stems and carbon loss occurred.

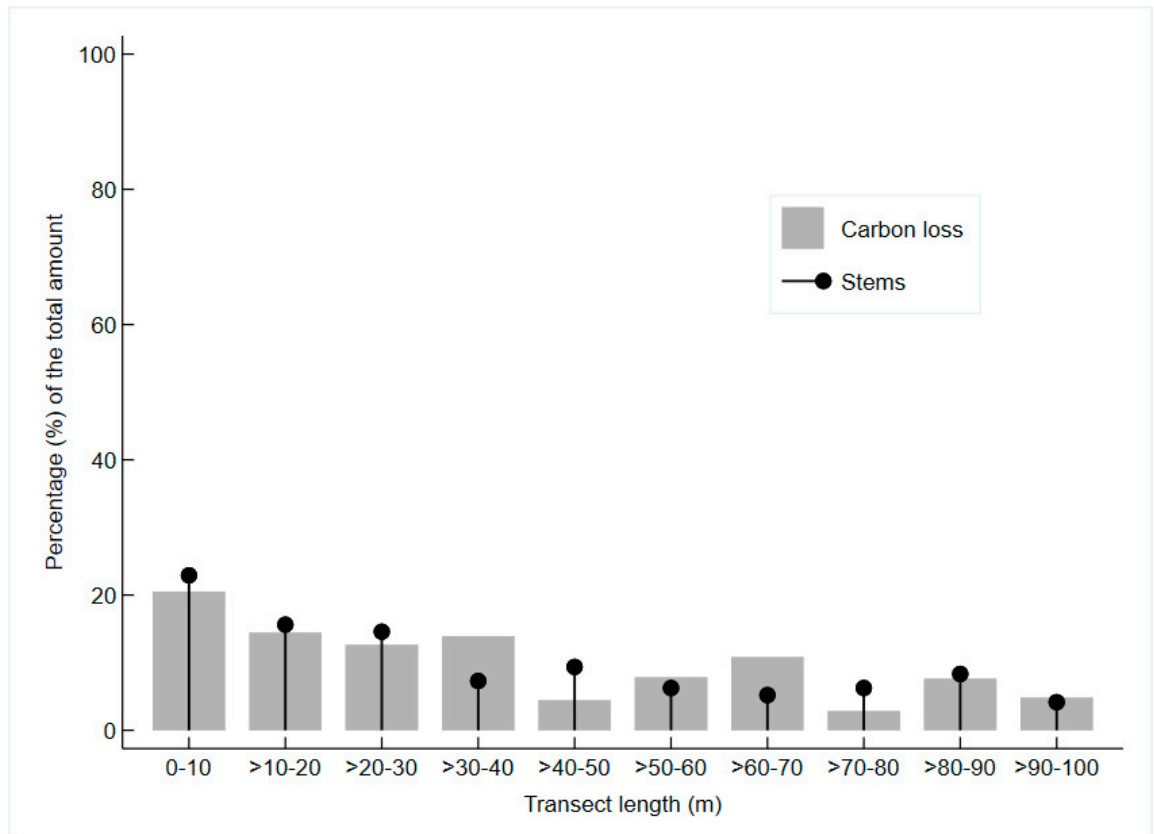

**Figure 4.** Distribution of stems and carbon loss from mining degradation damage at 10 m interval as a percent of all mining damaged trees and total carbon loss along the 100 m length of the fixed-area transects.

### 3.2. Emissions from Mining Degradation Compared to Deforestation Emissions

From 2011 through to 2014, an average of 10,663 ha was deforested annually across Guyana. The emission factor for deforestation in Guyana's national forest monitoring system was 1104 t $CO_2$ e ha$^{-1}$, resulting in annual average emissions of 11.8 million Mg $CO_2$ e. The average annual area of 100 m buffers around new mines between 2011 and 2014 was 40,250 ha. The average annual emissions applying the emission factor of 8 t $CO_2$ ha$^{-1}$ is 322 thousand t $CO_2$. This accounts for only 2.7% of carbon emissions from deforestation.

## 4. Discussion

This study and approach are designed to cost effectively assess forest edge degradation. The very simple method presented here creates emission factors for buffer areas around deforestation and can be applied globally (adjusting only the buffer width and emission factor to the specific circumstance) to determine greenhouse gas emissions from forest edge degradation, defined here as forest carbon losses in remaining forest adjacent to deforestation and occurring without a clearly apparent predictable pattern. The method uses existing remote sensing data for deforestation and ground data gathered at one point in time in a manner consistent with IPCC guidelines. It can be applied at low cost to evaluate the significance of forest edge degradation and to provide GHG emission estimates. The method is as simple as running transects out from recorded deforestation and calculating an emission factor for a given buffer width. This is appropriate to forests surrounding mining as in the case studied here, but has the potential to be applied to other forms of forest edge degradation, such as livestock incursion

into forests, the often overlooked degradation occurring at forest edges through microclimate changes, fuelwood collection, or localized entrance of anthropogenic fires into forests [45]. The specifics of the method, including the buffer width, the transect size, the measurements collected, and the allometric equations used should be adjusted for the specific geography and the driver of the forest degradation.

A cost-assessment has not been conducted given that a comprehensive high resolution analysis of forest degradation has not been implemented in Guyana by the government in an ongoing and sustainable manner due to the prohibitive costs this would incur. However, the cost difference is fully self-evident. A comprehensive high resolution approach would require annual or biennial purchase of high resolution remote sensing imagery with analyst time to define all losses in canopy cover that can be recorded as forest degradation, which would then be paired with emission factors to be associated with such losses. The field measurements, such as those conducted in the simplified method, are also needed for a comprehensive assessment and in addition to this there is a requirement for high resolution data on a spatially and temporally significant level with commensurate analyst time. As described by Goslee et al. in this journal [46], a pragmatic approach to emissions accounting, as detailed here in the case of forest edge degradation, results in substantial cost savings for insignificant emission sources.

Remote sensing techniques are commonly used to quantify forest degradation. Their calibration relies on the ground-based estimation of carbon emissions and removals [13,47,48] because quantification of forest degradation relies on the ability to relate degradation activities to disturbances in the canopy. Successful detection depends on the disturbance type, frequency, intensity, extent, and species composition [6]. Spatial and temporal resolutions of remotely sensed datasets are limiting factors to identifying disturbance on the ground. Even high spatial and temporal resolution imagery might not be able to capture minor forest structural disturbances [49]; thus, requiring ground-observations. Access to high-quality mapping, validation datasets, and ground measurements have been beneficial in estimating carbon emissions from forest degradation and in reducing uncertainty of estimates [50].

Emissions from forest degradation due to large-scale mining in Ghana were estimated at 2.1 Mg C ha$^{-1}$, including above and belowground carbon pools [51]. Using the IPCC default root:shoot ratio conversion factor (0.3), carbon emissions from aboveground biomass would be estimated at 1.5 Mg C ha$^{-1}$. The study of Amoako et al. [51] was based on ground-survey data covering 60 ha of mining degraded forests, with emission estimates in the range of those from the current study (2.2 Mg C ha$^{-1}$, at 95% CI 0.0–10.2 Mg C ha$^{-1}$). It should be noted, however, that details on the sampling design and on the steps followed in data generation steps were missing from the Amoako et al. study.

We are not aware of additional studies quantifying carbon emissions from mining degradation. The memorandum of understanding (MOU) between the governments of Norway and Guyana specified that the area of 500 m buffers around annual deforestation from mining be reported and that a 50% reduction in the carbon stock in these buffers would be taken due to degradation. Neither the area nor the stock reduction values were based on evidence. Remote sensing data, both from unpublished data and as reported here, show that there is clearly forest degradation associated with mining in Guyana, although the area impacted is much less than originally suggested in the MOU. Further underlining the invalidity of the 500 m transect, the number of stems damaged and the carbon loss in the damaged trees due to mining activities is higher within 0–50 m of mining deforestation than within 50–100 m. The field work and data analysis described here also indicate that the magnitude of loss in carbon stock due to mining activities in the 100 m buffers is considerably less than the 50% loss from the MOU—losses equal only around 1% of the aboveground tree pool.

Historical estimates of $CO_2$ emissions from deforestation, by all drivers, and from forest degradation by selective logging were made as part of Guyana's national REDD+ program [28]. The average annual emission estimates for the period 2011–2014 are shown in Table 5. Here, we show that forest degradation emissions associated with mining represent only 2% of total recorded forest emissions.

**Table 5.** Percentage of average annual (over period 2011–2014) greenhouse gas emissions by source in Guyana.

| Emission Source | Emissions (Million Mg $CO_2 \cdot yr^{-1}$) | Percentage of Total Emissions in Guyana |
| --- | --- | --- |
| Deforestation by all causes | 13.2 | 76% |
| Forest degradation by all causes [†] | 4.1 | 24% |
| *Deforestation from mining* | 10.7 | 62% |
| *Forest degradation from timber harvest* | 3.8 | 22% |
| *Forest degradation from mining* | **0.3** | **2%** |

[†] Forest degradation numbers from infrastructure only from 2012–2014

These estimates are likely an overestimate as the carbon impact of forest degradation presented here represents only the gross emissions and does not take into account how persistent the degradation might be or any regrowth and forest recovery—pioneer and small trees were observed in some of the year 2011 transects showing recovery of forest growth even while degradation was ongoing in other areas.

The approach of establishing 100 m buffers around deforested areas to arrive at emission estimates from associated forest edge degradation requires no additional image interpretation and can be conducted efficiently and at low cost. Given the low magnitude of Guyana's emissions resulting from mining forest degradation, this approach provides a sustainable and low-cost method for Guyana to conservatively include this source of emissions. In the absence of this new simplified approach, the likely only option would be to conduct an analysis with very high spatial resolution satellite or aerial imagery. Such an approach, although likely to provide a more accurate estimate of the area of forest degradation, is time and labor intensive, and our assessments have demonstrated that it is most often not possible to distinguish between natural tree mortality and that caused by mining-related activity. The buffer approach is conservative because even though the EFs estimated for the 100 m buffer are small, the buffer area will be large, and we have shown here that it encompasses a broad area with low risk of forest degradation.

## 5. Conclusions

It is clear that forest degradation surrounding mining represents a small source of emissions and a small percentage of emissions when scaled with other sources. As previously mentioned, the World Bank's Carbon Fund requires emissions from forest degradation to be reported when they represent more than 10% of forest-related emissions. Under this condition, the carbon emissions from degradation due to mining in Guyana would be considered insignificant under all circumstances.

This paper is expected to contribute in two ways to the global understanding and accounting of GHG emissions associated with forest degradation. First, this paper indicates that forest degradation associated with mining (and particularly alluvial mining excavation practices) is unlikely to be a globally significant source of GHG emissions that warrants fears of a major overlooked global emission. Second, the simplified methods in this paper can be used to derive country-specific estimates of carbon emissions from forest edge degradation including mining activities and other poorly understood forms of forest degradation to allow complete accounting at sufficiently low costs appropriate to the low net greenhouse gas emissions.

**Author Contributions:** Conceptualization, S.B., A.R.J.M., K.M.G., D.N.M.D. and P.W.; data curation, A.R.J.M., K.M.G. and H.S.; formal analysis, A.R.J.M., S.B., K.M.G., T.R.H.P. and H.S.; methodology, S.B., A.R.J.M., K.M.G., D.N.M.D. and P.W.; writing—original draft, S.B., A.R.J.M. and K.M.G.; writing—review and editing, K.M.G., T.R.H.P., and A.R.J.M. All authors have read and agreed to the published version of the manuscript.

**Funding:** This research has been funded by the Guyana Forestry Commission.

**Acknowledgments:** We would like to acknowledge the financial support from: (i) a contract between the Guyana Forestry Commission (GFC) and Winrock International (contract number GFC31/03/2015) and (ii) the REDD+ for the Guiana Shield project, coordinated by ONF International and funded by the European Commission, the French Global Environment Facility (FFEM), the French Guiana Region and the National Forests Office (ONF) of French Guiana. We thank all the hard work of the field crew, conducted primarily by staff from Winrock International (WI) and GFC, with assistance from the staff of Durham University (DU), and Indufor Asia Pacific (IAP). GFC, DU, and IAP were also very helpful in locating potential sampling sites and data collection methods. Special thanks to Towana Smartt (GFC), Rosa Rivas Palma (IAP), and Lara Murray (WI).

**Conflicts of Interest:** The authors declare no conflict of interest.

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
