# Peer review of "Accounting for Greenhouse Gas Emissions from Forest Edge Degradation: Gold Mining in Guyana as a Case Study"

_forests, doi:10.3390/f11121307_

Round 1

Reviewer 1 Report

The authors have attempted to address some of the shortcomings of the first version of the paper. They have resolutely contradicted my fundamental reservations about their approach. Certainly, it is not possible to conclusively clarify our different points of view. Let us therefore leave them as part of a scientific discourse.

There are still three main points of criticism:

  1. the method is not fully described The estimators are missing, both as regards the point estimators and the estimators of sampling errors.
  2. the study was conducted in a country with a low population density and one of the highest forest cover percentages in the world. It is inadmissible to generalize from this special case for the global level.
  3. in the whole paper the costs of the inventory are not considered. Therefore the basis for the assertion that a cost-efficient procedure is presented here is missing.

Detailed comments:

Line 59/60

Provide references

Line 62/63

Sentence incomplete

Line 74

There is something wrong with the syntax

Line 73 ff

The terminology is unfortunate. The authors use the term "concentrated degradation" to denote degradation in a delimited area, while the term "diffuse degradation" is used to denote degradation in the vicinity of deforestation. Since the area around a deforestation area can also be delimited (for example, by a buffer), the chosen definition is not clear. Intuitively, the terms diffuse and concentrated refer more to the spatial distribution of degradation, i.e. the spatial pattern of canopy gap sizes.

Line 146 ff., Fig 1

Replication of images

Line 156

It should be made clear that a buffer width of 100m only applies to the case study in Guyana. In other countries or in areas with higher population density, effects could also occur at distances of more than 100m.

In addition, the spatial impact is likely to depend on the cause of degradation (for potential causes see line 80 to 82)

Line 192 ff

In Fig. 2 there are rough points which show the position of the transects. It can be seen that often clusters of transects have been recorded (test sites 4, 5 and 2). Therefore the sample design here would be a combination of line transect sampling and cluster sampling. This would have to be taken into account when choosing the evaluation algorithms, i.e. the use of simple random sampling would not be appropriate in these test sites.

Statistical rational for a sample size of 41 transect should be given

Selecting the transects with respect to accessibility introduces bias. How was the bias corrected?

Line 215/216

How was the problem of buttress roots handled?

Line 226/227

How is the fact that transects were not selected statistically but arbitrarily (i.e. due to “accessibility and availability”) taken into account in the estimation procedures?

Line 231

Give more detail on the selection of biomass equations. Chave et al. present several approaches. How was the environmental factor selected? Why were equations from Chave et al. 2005 used? There is a more recent paper available presenting updated biomass equations, that should be used instead (Chave et al., 2014)

Line 237 ff

How was the model error introduced by estimating DBH from basal diameter measurements taken into account?

Line 248

The statistical approach used is very simple. Why was no generalized linear model used, which is state-of-the-art for such analyses? The analysis method used causes various problems, including the problem of multiple testing (no adjustment was made for the test for multiple endpoints, which is why the statements made are descriptive at best but not confirmatory).

Line 257 ff

Provide the number of standing trees (i.e. trees with DBH measurements) and the number of stumps (i.e. trees with basal diameter measurements and estimated DBH).

Line 262 ff

The estimation procedures are missing

Line 267 ff, Tab. 2

For understanding the meaning of the presented C-loss it is necessary to know the ABG C-stock of the respective forests. A paired comparison (transect level, test site level) would be helpful.

Line 274 ff, Table 3

The information is misleading. It is of less interest whether the intensity of degradation varies between years. What is of interest is whether the C-stock differs between control areas and the areas under consideration, i.e. comparing the C-stock changes due to degradation to the natural fluctuation of ABG-C-stocks.

Line 291 ff

The assessments have been conducted in different test sites. It would be helpful if you could present the data separately for each test site. Only then can it be excluded that an increased degradation in one site is not compensated by the development in another site. Again, a sound evaluation method would allow a more targeted analysis.

Line 307 ff

Degradation often has the character of a long-term effect, or the degradation effect occurs with a time delay. Therefore, the long-term degradation can be (significantly) higher than the current degradation. How is this phenomenon considered?

Line 320

Throughout the paper the costs of the assessment are not discussed. Therefore, it cannot be postulated that a cost-efficient method is presented. Especially the comparison to other (more cost efficient?) sampling design alternatives is missing.

Line 324

“applied at low cost” - see comment above

Line 319

Since the method is only incompletely described (among other things, the estimation procedures are missing), it is not comprehensible. Therefore a transfer to other areas is not possible.

Line 342

This paragraph illustrates the background of the study. Certainly, in the context of the MOU between Guyana and Norway, it is useful to check one or the other assumption. For the broad readership of Forests, however, the methodological aspects should be in the foreground. Here the present paper is too brief. 

Line 399 ff, Conclusions

The statement that on a global level degradation through mining does not play a significant role cannot be made in this generality. The study was conducted in a single country that is an outsider at least in terms of population density and forest cover.  

The conclusion regarding the applicability of the method is also not appropriate. The estimation procedures are not presented. They are not quite trivial, because purposive sampling is combined with cluster sampling and line-transect sampling. Furthermore, the clusters have unequal size. All this leads to relatively complex estimators, especially for sampling errors. The latter are indispensable for an operational accounting of emissions. Since no costs are considered, it cannot be claimed that this is a cost-efficient method. And last but not least, statements about the suitability of a certain inventory design can only be made if an optimization based on different design parameters and a cost comparison of different alternatives has been performed. In the literature there are numerous examples for the development and optimization of inventory designs, which should also be taken into account in the current application. 

Literature

Asner et al., 2013 is cited twice

Include the following literature source:

Chave, J., Réjou-Méchain, M., Búrquez, A., Chidumayo, E., Colgan, M.S., Delitti, W.B.C., Duque, A., Eid, T., Fearnside, P.M., Goodman, R.C., Henry, M., Martínez-Yrízar, A., Mugasha, W.A., Muller-Landau, H.C., Mencuccini, M., Nelson, B.W., Ngomanda, A., Nogueira, E.M., Ortiz-Malavassi, E., Pélissier, R., Ploton, P., Ryan, C.M., Saldarriaga, J.G., Vieilledent, G., 2014. Improved allometric models to estimate the aboveground biomass of tropical trees. Global Change Biology 20, 1365-2486.

Author Response

We appreciate the opportunity to respond to comments. Please see attachment for our detailed responses.

Reviewer 2 Report

The paper has been improved significantly.

There are some grammatical errors, I trust our production team can correct them.

Author Response

(The authors gave the same response as above.)

Reviewer 3 Report

The topic of the article is important and accessing forest degradation in mining areas make it more interesting and pertinent. Very few articles have address this linked when deforestation topics at mining areas are concerned. However, it is important to mention that this manuscript is not yet ready for publication. There is no clear definition of deforestation and forest degradation in this paper, it uses new terminologies like concentrated and diffuse degradation, term referring to location where degraded areas which can be concentrated and diffuse. The results obtained in this paper was always expected and the authors wrote themselves in the conclusion that forest degradation associated with mining is unlikely to be a globally significant source of GHG emissions. I recommend the authors to use satellite images and make a forest land cover change (year 2011-2013) analysis in the area where the transects were made and link it by the obtained results. Only them, the presented results can gain impact and balance and resubmit to the journal.   

Author Response

(The authors gave the same response as above.)

Round 2

Reviewer 1 Report

The authors have also tried to improve the manuscript in this new round. However, it now becomes obvious that they are not very familiar with the methods of inventory statistics and especially the optimization of forest inventories from the point of view of cost-effectiveness. An inventory design selected according to whatever criteria is adopted uncritically and transfer to other applications is recommended. It is not (anymore) appropriate to make a inventory design suggestion without discussing it in the contact of alternative designs.

It is not possible to judge the performance the method, because benchmarking by cost/efficiency with design alternatives is missing. Thus, the paper lacks a connection to the current discussion on selecting optimal forest inventory designs, which has progressed much further than recommending any inventory procedure for general application without comprehensible justification or verifiable indicators.

Finally, I am not sure whether the chosen text form does meet the intention of the authors. The contribution should serve to make an inventory idea by Sandra Brown, whom I appreciated very much, accessible to a broad audience. Unfortunately Sandra Brown has passed away. Whether in this situation an article with her name as first author is the right approach, I dare to doubt, but leave a decision to the editor. I would prefer if the authors would concentrate on a sound presentation of the inventory method, inventory statistics and cost-effciency optimization and publish this article in honor of Sandra. 

Reviewer 3 Report

I provided a detailed review on those particular points to be improved before publication. However, the author did not do as suggested and the answers were rather confrontational.  

I would like to recommend the authors to answer the questions rather than confront them. This paper has a potencial to be publish in the journal however, it need some polish on the introduction and methodology.

Here are the detailed from the first review. My assumption is that the authors did not see the detailed review: 

General comments

The Correspondence contact is not consistent. The contact email of author number 2, is different from the one stated as correspondent author.

Introduction

Line 61. Please define the abbreviation of MRV

Line 65. I recommended to replace the word describes by another that indicate a definition of forest degradation.

Line 67 -74. I would recommend to be very careful with utilization of new terminologies such as Concentrated degradation and Diffuse degradation. The arguments that justify the utilization of these terminologies have no balance in this paper. Forest degradation is already defined in many ways, the occurrence of degradation depends on the drives forces that impacts this same degradation therefore, it does not matter where degradation occur and how; it is simply forest degradation. Authors are recommended to rewrite these statements in another version to avoid conflicts in scientific terminologies.

Line 78. Which parts of the World?

Line 93- 95. Please remove: here mining represents diffuse forest degradation surrounding deforest such as livestock encroachment or the penetration of desiccating climate on deforestation edges. There is a literature review and its important to not present assumptions.

Line 103 – Again replace concentrated degradation with another know fashion terminology.

Line 119- Gold mining represents 88% of deforestation in Guyana? Could the authors mention some references that say so?

Line 123. It is important to be consistent of what you are presenting. Now you are presenting dispersed forest degradation and more the author are linking it to deforestation. Could the author draw a line inside this article the deference between forest degradation and deforestation?

Methodology

The characterization of the study area is missing.

Line 149. In table 1. Could please the authors define the of years in parentheses in Data products? Guyana forest map2011 (year 2, year 3 and year 4).

Line 163. What were the criteria for identification of those five arear? And how did the authors know that 41 transects at the sites could be sufficient for this study? How the transects were distributed along the five areas? What were the distances between the transects?

Line 195. In the tropics what? You mean in the tropic forest?

Line 210. Please include the reference for Chanve at al. equation.

Line 223. Please specify the concrete data you used non-parametric statical analyses – median, Wilcoxon rank-sun and Mann and Whitney U tests. Please describe it in the methodology.

Results

How can the authors justify that 3.8% trees have died due to mining activities?

Line 269. The frequency distribution of mining-damaged stems shows an expected gradual decline with an increase in distance from the edge of the deforestation. What could be the opposite scenario in here?

Line 272. How can the authors prove this statement in the results when the article did not access deforestation issues? I recommend to mix deforestation issues in here…

Line 278. This is completely of the context; the article does not address deforestation.

Discussion

Line 319. The discussion should be based on the results you obtained and the comparison with MOU between the Government of Norway and Guyana is not sustained as this MOU is not a scientific document.

References

Line 415. Please line up the reference.

Author Response

This manuscript is a resubmission of an earlier submission. The following is a list of the peer review reports and author responses from that submission.

Round 1

Reviewer 1 Report

This paper describes the carbon emissions generated around mining areas in Guyana.

In contrast to the claim made by the authors, no method is presented that is generally applicable to the assessment of emissions associated with mining.

The paper is primarily a narrative about emissions in the specific environment of mining areas in Guyana. Whether these findings are transferable to other regions remains to be proven. The authors do not provide evidence of transferability.

The method presented is not convincing either, as its development does not follow good practice. Estimation algorithms are not presented, and there is no information on the calculation of sampling errors (although these could be taken from standard literature). A cost analysis is not carried out, nor is a design optimization presented. Also missing is the comparison of alternative designs and the selection of the most cost-efficient approach.

Both the results and the method development of this paper show such large deficits that a publication cannot be recommended.

Reviewer 2 Report

There are several published papers which have addressed this issue all around the world. However, in Guyana, there are only few papers in this area but for many reasons this paper is not acceptable at this stage. My suggestions are as follows:

General suggestions:

  • Introduction
    • This paper is about forest degradation around mining areas. However, issues related to forests degradation in mining areas are not discussed. Please discuss this issue and possible reasons globally and make a case for your study. Then, you can discuss the same issue of Guyana and solve the problem taking case example of Guyana.
    • There are couple of issues in this para “The World Bank [2] under its Carbon Fund requires emissions from forest degradation to be accounted where ‘significant’ – defined as more than 10% of all forest-related emissions. However, it is unclear as to what activities cause significant forest degradation and how to measure and monitor such emissions cost-effectively when they are significant”. First this 10% rule is not only of WB. First, This is a common provision of many funding bodies. Discuss them. Second, the statement “it is not clear what activities cause significant forest degradation” is completely wrong. There are 1000s of research papers around this issue. As suggested above, rewrite it.
    • Some discussions are highly relevant in method section. Such as “We propose that this approach can be used to estimate the gross emissions from forest degrading activities related to gold mining by measuring damaged trees in the vicinity of mined areas. Because the intent is to focus strictly on carbon loss that results from human activity, it is critical that measurements are focused on trees (or stumps) whose mortality is caused by human impact, such as harvesting to build a mining camp or a trail, or mortality as a result of flooding or mine tailings”. Most of the discussion in this para goes to method section.
    • There are several contextual and grammatical issues throughout the paper. Moreover, there are several repetitions in the paper. A through revision focusing these issues are necessary.
  • Methods
    • Tables and figures are not self-explanatory. Please refine them.
    • Please provide reference for the data (Table 1)
    • Discuss the basis of selecting number of transects in different year.
    • In section 2.3, “basal diameter at 5 cm above ground were taken when DBH is not measurable. Discuss how you harmonise those two different dataset?
    • Provide reasons why height was not considered/measured.
  • Results:
    • In table 2, median values are skewed and therefore the curve in not normal. This has serious implication on your methodology, sample selection and findings. Moreover, implication of results is extremely limited. This is a major issue of this paper. Please explain these issues with satisfactory scientific logics. Also, you need to refine your discussion and conclusion sections to highlight these issues and limitation of your study.
    • Table 2, provide mean values as well in new column.
    • Table 3 and related sections, why you need to test this hypothesis. What are its implications. Discuss on method section.
    • Section 3.2 Emissions from mining degradation compared to deforestation emissions. There is not enough discussion/linkage with degradation. Please expand it
    •  
  • Conclusion
    • Write take home message from your own study. Do not cite or discuss papers in this section. Please move them into the discussion section.
    • Highlight weaknesses of this study and provide some suggestions for further study.

  • The Introduction and Discussion parts are very poor. There are so many literatures in this area but the authors have hardly discussed them. Please use following literatures both in the introduction and discussion sections.

Some of the key literatures:

  1. Pandey, SS. Maraseni, T.N., Cockfield, G. (2013). Major drivers of deforestation and forest degradation in developing countries and REDD+. International Journal of Forest Usufructs Management, 14 pp. 99-107
  2. Poudyal, B.H., Maraseni, T. N., Cockfield, G. (2019) Implications of selective harvesting of natural forests for forest product recovery and forest carbon emissions: Cases from Tarai Nepal and Queensland Australia, Forests, 10(8), 693; https://doi.org/10.3390/f10080693
  3. Paudyal, B.H., Maraseni, T. N., Cockfield, G. (2018) Evolutionary dynamics of selective logging in the tropics: A systematic review of impact studies and their effectiveness in sustainable forest management, Forest Ecology and Management 430, 166–175